# Identification of Genomic Characteristics and Selective Signals in a Du’an Goat Flock

**DOI:** 10.3390/ani10060994

**Published:** 2020-06-06

**Authors:** Qiuming Chen, Zihao Wang, Junli Sun, Yingfei Huang, Quratulain Hanif, Yuying Liao, Chuzhao Lei

**Affiliations:** 1Guangxi Key Laboratory of Livestock Genetic Improvement, Animal Husbandry Research Institute of Guangxi Zhuang Autonomous Region, Nanning 530001, China; cqm19860612@126.com (Q.C.); wangzh761006@163.com (Z.W.); sjn313@126.com (J.S.); Huangyingfei2020@outlook.com (Y.H.); 2Key Laboratory of Animal Genetics, Breeding and Reproduction of Shaanxi Province, College of Animal Science and Technology, Northwest A&F University, Yangling 712100, China; leichuzhao1118@nwafu.edu.cn; 3Computational Biology Laboratory, Agricultural Biotechnology Division, National Institute for Biotechnology and Genetic Engineering, Faisalabad 577, Pakistan; micro32uvas@gmail.com; 4Department of Biotechnology, Pakistan Institute of Engineering and Applied Sciences, Nilore, Islamabad 45650, Pakistan

**Keywords:** Du’an goat, whole genome sequencing, selective signals

## Abstract

**Simple Summary:**

Goat plays an irreplaceable role as domestic animals by providing milk, meat, and fiber. With the advancement in human civilization, over 500 goat breeds with various adaptive traits have been established. Whole-genome sequencing and subsequent analyses (including selective sweep analysis) are important tools to reveal the genetic basis underlying these adaptive traits. In this study, we investigated the sequenced genomes of 15 Du’an goats from one flock to identify the genomic characteristics and candidate genomic regions which might be involved in the adaptive traits. Selective sweep analysis revealed selected genes and/or pathways related to immune resistance, small body size, and heat tolerance. The identification of genomic characteristics and selective signals will not only help understand the demographic history and the genetic mechanism underlying adaptive traits but can provide novel insights into genetic diversity and development of breeding strategies in Du’an goats.

**Abstract:**

The Du’an goat is one of the most important farm animals in the Guangxi Autonomous Region of China, but the genetic basis underlying its adaptive traits has still not been investigated. Firstly, in this study, the genomes of 15 Du’an goats from a breeding farm were sequenced (mean depth: 9.50X) to analyze the patterns of genetic variation. A comparable diversity (17.3 million single nucleotide polymorphisms and 2.1 million indels) was observed to be associated with a lower runs of homozygosity-based inbreeding coefficient and smaller effective population size in comparison with other breeds. From selective sweep and gene set enrichment analyses, we revealed selective signals related to adaptive traits, including immune resistance (serpin cluster, *INFGR1*, *TLR2*, and immune-related pathways), body size (*HMGA2*, *LCOR*, *ESR1*, and cancer-related pathways) and heat tolerance (*MTOR*, *ABCG2*, *PDE10A*, and purine metabolism pathway). Our findings uncovered the unique diversity at the genomic level and will provide the opportunities for improvement of productivity in the Du’an goat.

## 1. Introduction

The goat *(Capra hircus*) was one of the oldest livestock species domesticated in the Zagros Mountains in the Fertile Crescent, around 10,000 years ago [1]. Domestic goats play an important role in the transition of human civilization from foraging and hunting to farming and herding by supplying milk, meat, and fiber. Due to natural and anthropogenic factors (e.g., climate changes, human migration, and population expansion), there are now more than 1006 million individuals and 557 registered breeds worldwide [2,3].

The Du’an goat is a small-statured goat breed which is mainly reared in Du’an county of Guangxi Zhuang Autonomous Region in China and is sporadically found in the neighboring counties. The number of Du’an goats raised per year is more than 0.6 million. Du’an goats are featured as a meat-type breed with the following characteristics: (1) small body size with the average body weight of 41 kg in the female; (2) high heat tolerance with average temperatures of 29 Celsius degree in July in the Du’an county; (3) high disease resistance against *Fasciola hepatica*; (4) diverse types of coat color (white, black, etc.); (5) tender meat; and (6) diverse types of horn (Figure 1) [4,5]. However, the genetic basis underlying these characteristics is still not investigated.

The decreasing cost of high-throughput sequencing has resulted in considerable progress in genome characterization and demographic history of various organism. Moreover, the selection sweep analysis from the genome data set helped in the identification of the candidate genes/regions associated with adaptive traits. For example, a whole-genome sequencing study showed a very high diversity associated with linkage disequilibrium in Moroccan goat [6]. The selection sweep analysis of Korean indigenous goats revealed selective signals for Salmonella infection and cardiomyopathy [2]. One of the recently-published studies including 164 modern domestic goat, 52 ancient goat, 24 modern bezoar, and 4 ancient bezoar genomes indicated that Asian goats are genetically distinct from European and African samples. More importantly, the study identified a strong signature of selection harboring *MUC6* gene conferred enhanced immune resistance to gastrointestinal pathogens [7]. Furthermore, studies showed selective signals in other important traits, such as hair growth in the Cashmere goat [8], litter size traits in the Laoshan dairy goat [9], and high altitude adaptation in the Tibetan goat [3]. However, there have been scarce or no attempts to analyze the genomic characteristics and selective signals in the Du’an goat.

In the present study, we performed whole-genome resequencing of 15 Du’an goats to identify genomic characteristics and selective signals. The revelation of selective signals related to immune system, body size, and heat resistance will help further investigate the genetic mechanism underlying adaptive traits in the domestic goats of South China. Our findings will also provide valuable resources for breeding programs and managemental strategies regarding the Du’an goat.

## 2. Materials and Methods

### 2.1. Animal Sampling

The ear tissue of 15 Du’an goats was collected from a designated breeding farm in Du’an county of Guangxi Zhuang Autonomous Region, China. Genomic DNA was extracted from the ear tissue using the standard phenol–chloroform protocol [10]. The sampling procedure was approved by the Institutional Animal Care and Use Committee of Northwest A&F University (permit number: NWAFAC1019).

### 2.2. Genome Sequencing and Variant Calling

Each DNA sample was used to construct paired-end sequencing libraries with the insert size of 350 bp. The constructed libraries were sequenced using the Illumina NovaSeq 6000 platform at the Novogene Bioinformatics Institute, Beijing, China. The length of reads was 150 bp. The raw reads were filtered using Trimmomatic [11] with the following parameters: LEADING:20, TRAILING:20, SLIDINGWINDOW:3:15, AVGQUAL:20, MINLEN:35, and TOPHRED33. The clean reads were aligned to the goat reference assembly (ARS1) [12] using Burrows-Wheeler Alignment Maximal Extract Matches algorithm [13] with default parameters. After alignment, the short reads in the BAM file were sorted and duplicated reads were marked (Appendix A).

For SNP (single nucleotide polymorphism) and indel detection, we orderly used the modules (HaplotypeCaller, CombineGVCFs, GenotypeGVCFs, and SelectVariants) of Genome Analysis Toolkit 3.8 (GATK) [14] to call the raw variants (Appendix A). The raw SNPs were filtered using the VariantFiltration module of GATK with the following parameters recommended by GATK: SOR > 3.0, QD < 2.0, FS > 60.0, MQ < 40.0, MQRankSum < −12.5, ReadPosRankSum < −8.0, DP < 47, and DP > 427. Similarly, raw indels were filtered using the following parameters recommended by GATK: QD < 2.0, FS > 200, and ReadPosRankSum < −20.0. The package ANNOVAR [15] was used to annotate the variants to identify the protein coding mutations caused.

### 2.3. Population Genetic Analysis

To check for the relatedness among the samples, identity-by-descent testing was performed using PLINK (--genome) [16]. Nucleotide diversity (--window-pi 50,000 --window-pi-step 50,000) and inbreeding coefficient (--het) were also calculated to detect the genomic characteristics in the Du’an goat using VCFtools [17]. The linkage disequilibrium (LD) decay in Du’an goats was estimated using PopLDdecay [18].

Runs of homozygosity (ROH) in the Du’an goat was detected using PLINK [16] (Appendix A). The ROH-based inbreeding coefficient (*F*_ROH_) was calculated as the average fraction of the autosome covered by ROH. In addition, effective population size (Ne) was calculated using SNeP version 1.1 with default parameters [19].

### 2.4. Detecting Positive Selection

Before detecting the selective signals, one of the samples was removed due to relatedness (identity-by-descent PI_HAT = 0.29). Evidence of positive selection was investigated through three statistics. First, the SNPs were filtered with minor allele frequency < 0.05 and missing rate > 0.10 using VCFtools [16], and missing alleles imputation and haplotype inference were performed using Beagle version 4.1 (Appendix A) [20]. The integrated haplotype score (iHS) was estimated from haplotype information to find the selected alleles segregating at intermediate frequency in the Du’an goat using selscan [21]. We also normalized the iHS score using the norm module of selscan. For the iHS statistic, the fraction of SNPs with |normalized iHS score| > 2 in 50 kb windows with 20 kb increments was calculated across the autosomes.

Second, based on the empirical frequency spectrum with all allele frequencies across the autosomes, sweepfinder2 [22] was used to calculate the composite likelihood ratio (CLR) statistics for sites every 1 kb (−lg 1000 FreqFile SpectFile OutFile) using allele frequency information to detect the completed sweep. In order to define candidate regions, the genome was divided into 50 kb windows with 20 kb increments. In each window, the maximum CLR was defined as the test statistic.

Third, the nucleotide diversity (θπ) was calculated (--window-pi 50,000 --window-pi-step 20,000) using VCFtools [16]. Outlier regions supported by two or three methods (higher iHS: top 1%, higher CLR: top 1%, and lower θπ: bottom 1%) were defined as the candidate regions under positive selection. In addition, the Tajima’D statistic was also calculated (--TajimaD 20,000) across the autosomes to consolidate our results using VCFtools [16].

### 2.5. Candidate Genes Analysis

Based on the goat reference genome (ARS1), a custom perl script was used to annotate the regions under positive selection. The protein-coding genes overlapped with the regions under positive selection were defined as candidate genes. Kyoto Encyclopaedia of Genes and Genomes (KEGG) pathway analysis was performed on the candidate genes using KOBAS 3.0 [23]. Fisher’s exact test with false discovery rate testing was executed and a corrected *p*-value of less than 0.05 was chosen as an inclusion criterion for functional categories.

## 3. Results

### 3.1. Sequencing and Identification of SNPs and Indels

To identify the SNPs and indels in the Du’an goat, we sequenced 15 genomes. In total, ~2.81 billion clean reads were generated and aligned to the goat reference genome (ARS1). The average genome coverage was 99.68% (ranging from 99.24% to 99.79%) with an in-depth mapping coverage of 9.50 folds (ranging from 7.74 to 11.47) (Appendix A).

After quality filtering, a total of 17,317,364 SNPs was retained, while the average number of singletons, observed homozygosity, and observed heterozygosity were 163,581 (ranging from 96,181 to 373,907), 11,589,894 (ranging from 10,995,608 to 12,573,915) and 4,433,699 (ranging from 3,414,574 to 5,114,509), respectively. The transitions/transversions ratio was 2.3359, which was comparable to that in the Dazu black goat (2.3347) [24], and lower than the Moroccan goat (2.3552) [6], suggesting that the variant calling of the samples was done correctly. The majority of SNPs were identified in the intergenic (61.6%) and intronic (35.6%) regions, while, only a fraction of SNPs were detected in the genic regions including exonic (0.8%) and untranslated regions (0.7%). Among the exonic SNPs, 45,410 SNPs caused amino acid change, while 4009 and 83 SNPs caused the creation and elimination of a stop codon, respectively (Figure 2a). Similarly, 2,111,567 indels were detected, where, 0.3% were detected in exonic regions, 37.6% in intronic regions, 59.5% in intergenic regions, and 1% in untranslated regions. Among exonic indels, 1265 and 1020 indels led to frameshift deletion and insertion, respectively, and 58 and 4 indels led to the creation and elimination of a stop codon, respectively (Figure 2b).

We also investigated the genomic characteristics in the Du’an goat. According to identity-by-descent analysis, only one pair from the individuals showed PI_HAT value of greater than 0.25 (which was later excluded from the selection sweep analysis). The mean and median of nucleotide diversity were 0.0019 and 0.0017, respectively, in non-overlapping windows of 50 kb across the autosomes (Figure 2c). The average SNP-based inbreeding coefficient was 0.06613 in the Du’an goat (Appendix A). Linkage disequilibrium (LD) analysis indicated the physical distance between SNPs to be 77 kb (reported as half of its maximum) (mean r^2^ = 0.35) (Figure 2d), while, the average ROH coverage was 6.217 Mb (ranging from ~1–44.312 Mb). Accordingly, the average *F*_ROH_ was 0.0025 (ranging from ~0.0004–0.0179) (Appendix A). In addition, an estimated decrease in the Ne was also observed in the Du’an goat (from 2668 at 999 generations ago to 55 at 13 generations ago) (Figure 2e) (Appendix A).

### 3.2. Identification of Selective Sweep

To identify candidate regions and genes under selection in the Du’an goat, we used iHS statistic, CLR statistic, and nucleotide diversity. The regions with support from two or three methods (lower nucleotide diversity, higher iHS statistic, and CLR statistic) were defined as the candidate regions under selection (Figure 3a–c). After merging the overlapping regions, 124 candidate regions with 245 genes were identified under selection in the Du’an goat (Appendix A). Based on annotation of the ANNOVAR software, we also identified 131 nonsynonymous SNPs with the high frequency (>0.8) in 68 selected genes (Appendix A) and eight exonic indels with the high frequency (>0.8) in six selected genes (Appendix A).

An extreme iHS score was located in the serpin cluster. The selective signals in the serpin cluster were further confirmed by a higher value of CLR, local reduced nucleotide diversity, and lower value of Tajima’s D (Figure 3d). *HMGA2* was identified as a candidate gene, which showed a strong CLR signal and reduced nucleotide diversity. The region also showed a negative value of Tajima’s D indicative of an increase of rare alleles, such as recent bottleneck followed by expansion (Figure 3e). In addition, we found *MTOR* fell in a region that contained some of the most extreme CLR statistic and nucleotide diversity. The negative value of Tajima’s D also supported the selection in the region (Figure 3f). Other interesting candidate genes overlapped with selective signals were *IFNGR1*, *TLR2*, *ESR1*, *LCOR*, *ABCG2*, and *PDE10A*.

### 3.3. Kegg Pathway Analyses of the Candidate Genes under Selection

In order to provide more functional information about the candidate genes under selection, gene set enrichment analysis of the KEGG pathway was performed, resulting in nine significant pathways. The most significant pathway was olfactory transduction. Other pathways included four immune-related categories (amoebiasis, acute myeloid leukemia, staphylococcus aureus infection, and tuberculosis), two cancer-related categories (PD-L1 expression and PD-1 checkpoint pathway in cancer, and proteoglycans in cancer), and one metabolism-related category (purine metabolism) (Table 1).

## 4. Discussion

### 4.1. Genetic Diversity in the Du’an Goat

Study of genetic diversity plays an important role in the conservation and utilization of germplasm resources, revelation of evolutionary history, and investigation of phylogenetic relationships. The number of the detected SNPs in the Du’an goat (~17M) was lower than that in the Moroccan (~33M) and Korean indigenous (~37M), and higher than that in the Tibetan Cashmere (~12M), Chengdu Brown (~10M), and Jintang Black goat (~12M) [8], which could be explained by the difference in the number of individuals, reference genome, detection method, filtering criteria, or breed. The average nucleotide diversity in the Du’an goat was lower than that in the Iranian indigenous goat, while it was found to be higher than that in the Nubian and Korean indigenous goat. The average SNP-based inbreeding coefficient in the Du’an goat was higher than the Nubian and Korean indigenous goat. The average r^2^ in 50 Kb in the Du’an goat was higher than that in the Iranian indigenous, Moroccan indigenous, and Boer goat, while it was lower than that in the Nubian goat (Table 2). However, ROH-based inbreeding coefficient (0.0025) was lower in the Du’an goat than the Chengdu Brown (0.194), Tibetan Cashmere (0.068), Moroccan indigenous (0.085), and Jintang Black goat (0.119) [8]. The revelation of individual ROH may help the conservation of this breed, since animals with high levels of ROH could be excluded or assigned lower priority for mating to minimize the loss in genetic diversity. In addition, a decreasing Ne was observed in the Du’an goat, and the estimated Ne value was 55 at 13 generations ago, which was lower than that in the Arbas Cashmere (95) and Guangfeng goat (64) [25]. Similarly, a previous study also found higher Ne values in the Argentinian (67), French (57) and South Afrcian goat (93) at 10 generations ago [26], which supports our results. A recent decrease in Ne can be due to a stronger selective pressure in recent generations or to a recent bottleneck. From the above results, we can conclude that the Du’an goat has higher SNP-based inbreeding, lower ROH-based inbreeding, fewer variants, higher linkage decay, and smaller effective population size and nucleotide diversity comparable to the other locally-adapted populations, suggesting the unique genetic characteristics.

### 4.2. Putatively-Selected Genes

#### 4.2.1. Immune System

High disease resistance is one of the key features in the Du’an goat. A previous study reported that *Fasciola hepatica* is endemic in Guangxi Zhuang Autonomous Region inhabited by the Du’an goat [5]. Among the putatively-selected genes, serpins are serine proteinase inhibitors involved in host defense pathways [27]. An in-depth proteomic analysis of *Fasciola hepatica* intra-mammalian stages showed that five serpins were identified in adult excretory/secretory product and somatic soluble newly excysted juveniles, suggesting that serpins could be important for *Fasciola hepatica* establishment and survival in the host [28]. Other noteworthy genes in our candidate list were *IFNGR1* involved in hepatitis B virus and mycobacteria infection [29,30] and *TLR2* implicated in pathogen recognition and activation of innate immunity [31]. Interestingly, the ADAPTmap consortium found that the *IFNGR1* was one of candidate genes involved in environmental adaptation; another 10 candidate genes (*ARL8A*, *CDC25A*, *GPR37L1*, *IL22RA2*, *IQCE*, *KIT*, *LOC102181444*, *MAP28*, *NAV1*, and *PTPN71*) identified by the ADAPTmap consortium were also within the region under selection in the Du’an goat [32]. At the same time, we found four nonsynonymous SNPs with the high frequency in three immune-related genes in the Du’an goat (*IFNGR1*: c.A733G, p.I245V; *SERPINB7*: c.A82G, p.M28V; *TLR2*: c.T47C, p.V16A; and *TLR2*: c.A739G, p.I247V). In addition, gene set enrichment analysis showed four immune-related KEGG pathways. We could speculate that these genes, nonsynonymous SNPs, or pathways could have been affected by selection targeting at immune traits such as resistance to viruses, bacteria, or parasites.

#### 4.2.2. Body Size

Another important Du’an goat characteristic is its small body size. The average body weight is 48 kg in the adult male and 41 kg in the adult female. One interesting observation was the presence of *HMGA2* among the genes with strong signal of selection. *HMGA2* is a widely-studied gene explaining the difference in body size in many organisms including humans [33], cattle [34], dog [35], and horse [36]. Other putatively-selected candidates included *LCOR* associated with cattle stature [34] and *ESR1* associated with body height in two Swedish populations [37]. It is worth mentioning that there were two nonsynonymous SNPs with the high *LCOR* frequency in the Du’an goat (c.G3029A:p.G1010E and c.G2513C:p.S838T). In addition, we observed that two cancer-related KEGG pathways were detected in gene set enrichment analysis. In fact, the relationship between cancer and body size or obesity has been established by numerous studies [38,39,40]. From the above results, we can conclude that these genes or pathways were strong candidates contributing to the small body size in the Du’an goat.

#### 4.2.3. Heat Tolerance

Guangxi Zhuang Autonomous Region, inhabited by the Du’an goat, is in the south subtropical zone, where the average annual temperature is 17–23 Celsius degree, the average maximum temperature is about 29 Celsius degree in July [41]. In the genome-wide screen, we found a very strong selective signal with a high-frequent nonsynonymous SNPs in *MTOR* (c.G5507C:p.G1836A). A previous study has shown that *MTOR* is essential for the proteotoxic stress response, *HSF1* activation, and heat shock protein synthesis [42]. Moreover, heat stress experiments showed enhanced *MTOR* signaling in human and rat skeletal muscle [43,44]. A prominent study showed that *MTOR* could regulate phase separation of PGL granules to modulate their autophagic degradation and heat stress adaptation during *Caenorhabditis elegans* embryogenesis [45]. In fact, another study has proved that mutation of *MTOR* gene could be associated with heat tolerance in Chinese cattle [46]. Other strong selection candidates included *ABCG2* implicated in oxidative stress [47], and *PDE10A* implicated in a thermoregulatory role [48]. It has been demonstrated that the duration of thermal stress affects the expression of *ABCG2* in broilers [49]. In addition, we found that purine metabolism was significantly enriched in putatively-selected genes using KEGG pathways. A review has presented the function of heat shock protein and metabolic responses under heat stress or high ambient temperature [50]. Some studies have shown that purine metabolism is involved in heat stress [51,52]. All the above findings indicate that these genes or pathways may be important factors in Du’an goat adaptation to a subtropical environment.

## 5. Conclusions

Our study used resequencing data from the Du’an goat to identify genomic characteristics and selective signals. The descriptions of genomic characteristics (nucleotide diversity, SNPs, indels, inbreeding coefficient, linkage disequilibrium, ROH, and effective population size) are not only crucial for maintaining herd genetic diversity and development of breeding programs, but also help understand the demographic history in the Du’an goat. The revelation of selective signals related to immune system, body size, and heat tolerance will provide an important opportunity for further investigation of the genetic mechanisms and underlying adaptive traits in the Du’an goat. The identification of some candidate variants will facilitate investigation into related phenotypical data to detect causative variants. Further studies are hereby desired with broader and thorough investigation, leading towards more elaborative genomic characterization and demographic history based on larger datasets, augmenting the current study for better managemental and breeding policies.

## Figures and Tables

**Figure 1 animals-10-00994-f001:**
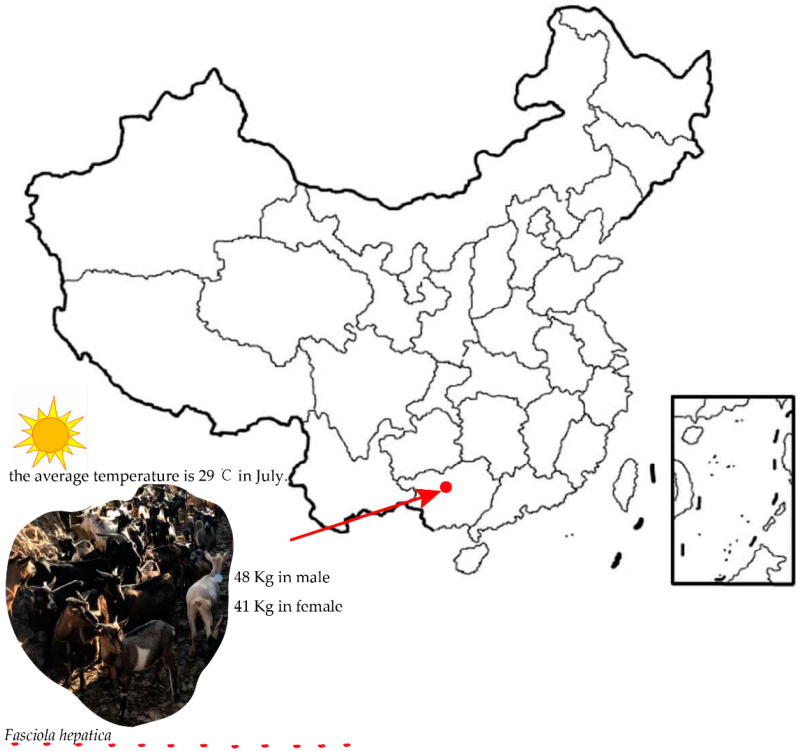
Breed characteristics and geographic location of the Du’an goat.

**Figure 2 animals-10-00994-f002:**
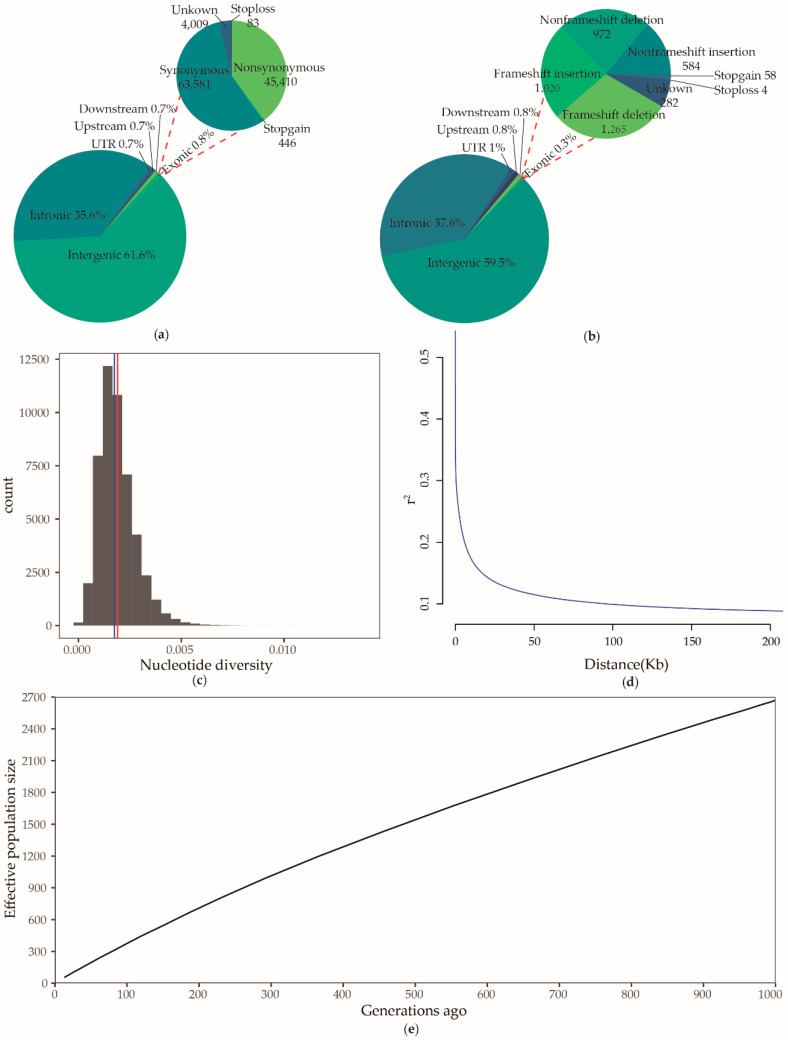
Identification of genomic characteristics. (**a**) Functional classification of the detected single nucleotide polymorphisms. (**b**) Functional classification of the detected indels. (**c**) Distribution of nucleotide diversity in non-overlapping windows of 50 kb across the autosomes. The red line and blue line indicate mean and median, respectively. (**d**) Genome-wide average linkage disequilibrium (LD) decay in the Du’an goat. (**e**) Effective population size over the past 1000 generations.

**Figure 3 animals-10-00994-f003:**
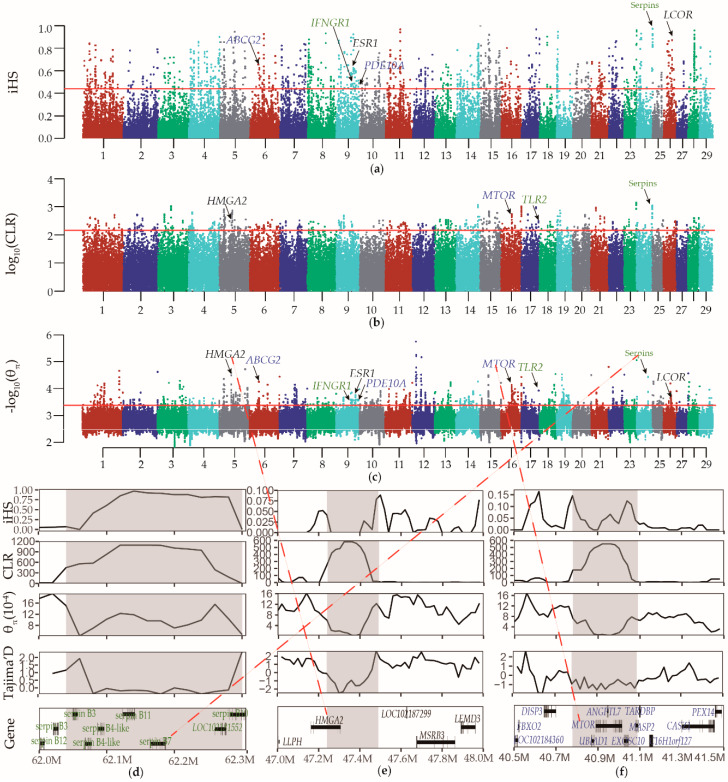
Identification of selective signals in the Du’an goat. Distribution of iHS statistic (**a**), composite likelihood ratio (CLR) statistic (**b**), and nucleotide diversity (**c**) in a 50 kb sliding window with a 20 kb step across all autosomes. The red lines indicate the corresponding threshold (iHS statistic > 0.4424167, CLR statistic > 148.2987 and nucleotide diversity < 0.0004230301). (**d**) Selective sweep at the serpin cluster on chromosome 24 (62.04–62.33 Mb) (gray area). (**e**) Selective sweep at the *HMGA2* on chromosome 5 (47.24–47.49 Mb) (gray area). (annotation, black rectangles and black lines represent exons and introns, respectively. (**f**) Selective sweep at the *MTOR* on chromosome 16 (40.78–41.09 Mb) (gray area). In gene.

**Table 1 animals-10-00994-t001:** Enriched gene ontology terms among the candidate genes under selection ^1^.

KEGG Term	Corrected *p*	Gene Count
Olfactory transduction	2.22 × 10^34^	44 (448)
Amoebiasis	0.00194	6 (95)
PD-L1 expression and PD-1 checkpoint pathway in cancer	0.0097	5 (89)
Acute myeloid leukemia	0.021178	4 (66)
Staphylococcus aureus infection	0.003584	5 (68)
Tuberculosis	0.023397	6 (179)
Purine metabolism	0.029284	5 (130)
Proteoglycans in cancer	0.032434	6 (203)
Salivary secretion	0.040032	4 (90)

^1^ Enriched terms are color-coded to reflect relatedness in functional proximity. Blue, immune system; green, cancer; and orange, metabolism. For each term, gene count shows number of candidate genes (total number of annotated genes).

**Table 2 animals-10-00994-t002:** Difference in genomic characteristics between the Du’an goat and seven other goat breeds.

Breed	Number of SNPs	Nucleotide Diversity	Inbreeding Coefficient ^1^	Average r^2^ (50 kb)
Du’an goat	17,317,364	0.001905	0.06613	0.3674
Iranian indigenous goat ^2^	35,742,191	0.001998	0.06229	0.087908
Moroccan indigenous goat ^2^	32,914,220	0.001859	0.06143	0.079364
Saanen goat ^2^	36,845,217	0.001783	0.00207	0.063634
Nubian goat ^2^	23,726,534	0.001117	−0.0234	0.487128
Boer goat ^2^	32,384,827	0.001724	−0.04455	0.091387
Korean indigenous goat ^2^	37,715,208	0.001472	0.01661	0.161031
*Capra aegagrus* ^2^	39,222,625	0.001804	0.06821	0.087908

^1^ Inbreeding coefficient was estimated using a method of moments. ^2^ The genomic characteristics in these breeds was identified by Badr Benjelloun et al. in 2019 [2].

## Data Availability

The sequencing reads of each sample have been deposited at GenBank with the Bioproject accession number PRJNA607828.

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
