# Peer review of "Identification of Genomic Characteristics and Selective Signals in a Du’an Goat Flock"

_animals, 2020, doi:10.3390/ani10060994_

Round 1
Reviewer 1 Report
The study performed genome resequencingof 15 individuals of Duan goat,and analyzed the genettic divesity and selcetvie ssweep .However, the size of one goat breed is small for selcetvie signals analysis, which will affect the accuracy of the result. THE study shrew light on the breeding and utilization of DUan goat in future.
1, In the abtract, the author need provide detailed information about genetic diversity of Duan goat, which can demostrate the level of genetic diversity.
2,In the abtract, resequencing depenth shoulde be listed.
3,few english wirting need futher to be improved.
4,IN material and methord,the relationship of 15 samples need to be shown, ,These samples aare unrelatid or other.....which is important.
Author Response
The study performed genome resequencingof 15 individuals of Duan goat,and analyzed the genettic divesity and selcetvie ssweep .However, the size of one goat breed is small for selcetvie signals analysis, which will affect the accuracy of the result. THE study shrew light on the breeding and utilization of DUan goat in future.
Response: Thank you for your careful comments. We clarified all critical points suggested by you.
1, In the abtract, the author need provide detailed information about genetic diversity of Duan goat, which can demostrate the level of genetic diversity.
Response: Thank you for your advice. We have provide detailed information (lines 25-26).
2,In the abtract, resequencing depenth shoulde be listed.
Response: Thank you for your advice. We have added the information (line 24).
3,few english wirting need futher to be improved.
Response: Thank you for your advice. We have checked English writing again.
4,IN material and methord,the relationship of 15 samples need to be shown, ,These samples aare unrelatid or other.....which is important.
Response: Thank you for your advice. We have added the information (line 72).

Reviewer 2 Report
Comment to the author
The manuscript by Chen Q et al. investigates the diversity and genomic characteristics of the Du’an goat breed, an autochthonous population of the Chinese Du’an county through whole genome resequencing. The authors analyse some diversity metrics and compare them with previous studies which performed similar analyses, and ultimately identify selection signatures that might be associated with some characteristics of the population.
The manuscript would greatly benefit from an English revision to fix the multiple typos, unclear and/or poorly connected sentences. Despite the insight provided by the investigation of a local breed with such a large number of individuals (>600K individuals), the paper presents several critical points that need to be addressed prior consideration for publication, including a proper description of the population, a deeper and more accurate investigation of the selection signatures and draw conclusions that are in contrast with the data provided. In particular, the authors should spend more times investigating single variants, looking for missense within the genes identified as under selection.
Below, you can find a detailed list of the comments to the authors.
Major revision
- Introduction
- Rows 41-44: Both the references provided are in Chinese, and some of them are difficult to access. Please provide alternative references or, if not possible, provide a better description of the population with pictures showing the phenotypes described.
- Material and methods
- Rows 69-71: please add some more details about the sampling: which criteria has been used to define the samples? Have they been chosen as unrelated? Are they representative of the whole area the Du’an county, or just of a small portion of it? Has any phenotype been measured?
- Rows 76-88: please specify the commands used to generate the dataset as a separate document, in order to guarantee the reproducibility of the analyses, or use alternative ways to share the code used (e.g. github, a readme file with all the details, R markdown etc.)
- Rows 82-83: has the genotyping been performed as a joint call or individual by individual and then the dataset been merged?
- Rows 84-87: how did the authors defined the filtering criteria? The hard filtering can vary greatly depending on the dataset. In absence of a truth dataset to use to train a VQSR model, it is better to assess multiple filtering values checking how the transition/transversion rate change in order to define the best set of filtering to adopt, and minimise false positives (retained) and negatives (excluded).
- In addition to the analyses performed, I would suggest to describe the number of singleton per individual, heterozygosity per individual, the Ne for the population at different generations (through SNeP or NeESTIMATOR v.2 for example) and the FROH. These additional metrics might provide additional insight into the demographic of the population, including bottlenecks and regions of unexpectedly high homozygosity.
- Rows 109-110: why not considering also the Tajima’s D for the selection of the signals?
- Results and discussion
- Rows 126-127: a coverage of 9.5 is not elevated. Please remove the adjective.
- Row 128: number of singletons? Ts/Tv ratio for the variants selected?
- Figure 1a and 1b: the two figures have the same caption.
- Rows 151-152: how did you defined a consecutive region? Simply if they overlapped or if they were within a certain distance? Please add some more details.
- Rows 195-198: the conclusion drawn by the authors is in clear contrast with the previous assumptions. If the population has higher inbreeding, less variants than the other breeds described, higher linkage decay and nucleotide diversity comparable to the other populations just means that the population has an unusually high level of homozygosity, with very few diverse haplotypes consistent with recent bottleneck. The addition of FROH, effective population size and Heterozygosity might provide better insight and allow for a deeper understanding of the population genetic diversity.
- The authors describe a series of genes under candidate selection in this population, and do not go any farther than this, despite having annotated the variants’ effect using ANNOVAR. It would be good to know whether there is any non-synonymous, high impact variants in the genes of interest.
- How does the variants relate with other studies performed in other goat breeds? For example, the ADAPTmap consortium studied selection signatures in multiple populations worldwide, including small ones. It would be good to compare the results identified with the results produced in these studies.
- Conclusions
- The authors have the opportunity to already detect candidate variants that, in the next studies, could be investigated jointly with phenotypical data to detect causative variants. I think this is worth further investigation, and to be specified here too.
Minor Revision
- Rows 12: replace cashmere with fibre.
- Rows 96-97: specify the parameters used to perform the imputation. Have the authors retained the physically phased genotypes (PGT field in the genotype annotation by GATK) prior imputation? This because Beagle doesn’t recognise the PGT field automatically, and therefore ignores the phased genotypes. Since read-based phasing is much more reliable than statistical phasing, and therefore it is wise to retain it.
- Rows 100-101: how did the author defined the ancestral alleles?
- Rows 159-150: please provide a bit more of explanation of the meaning of the Tajima’s D negative values (excess of rare alleles, derived by recent bottleneck or selective sweep) either here or in the materials and methods
- Row 206: replace dense with defence.
- Supplementary tables: I would suggest to break the tables in multiple, smaller pages. Not everyone has access to a A3 printer, and print these tables in A4 would make them hard to read
Author Response
The manuscript by Chen Q et al. investigates the diversity and genomic characteristics of the Du’an goat breed, an autochthonous population of the Chinese Du’an county through whole genome resequencing. The authors analyse some diversity metrics and compare them with previous studies which performed similar analyses, and ultimately identify selection signatures that might be associated with some characteristics of the population.
The manuscript would greatly benefit from an English revision to fix the multiple typos, unclear and/or poorly connected sentences. Despite the insight provided by the investigation of a local breed with such a large number of individuals (>600K individuals), the paper presents several critical points that need to be addressed prior consideration for publication, including a proper description of the population, a deeper and more accurate investigation of the selection signatures and draw conclusions that are in contrast with the data provided. In particular, the authors should spend more times investigating single variants, looking for missense within the genes identified as under selection.
Response: Thank you for your careful comments. We clarified all critical points suggested by you.
Below, you can find a detailed list of the comments to the authors.
Major revision
Introduction
Rows 41-44: Both the references provided are in Chinese, and some of them are difficult to access. Please provide alternative references or, if not possible, provide a better description of the population with pictures showing the phenotypes described.
Response: Thank you for your advice. We have added the picture (Figure 1).
Material and methods
Rows 69-71: please add some more details about the sampling: which criteria has been used to define the samples? Have they been chosen as unrelated? Are they representative of the whole area the Du’an county, or just of a small portion of it? Has any phenotype been measured?
Response: Thank you for your advice. We have added the information (line 72).
Rows 76-88: please specify the commands used to generate the dataset as a separate document, in order to guarantee the reproducibility of the analyses, or use alternative ways to share the code used (e.g. github, a readme file with all the details, R markdown etc.)
Response: Thank you for your advice. We have added all critical parameters in the manuscript (lines 82+85+88+89).
Rows 82-83: has the genotyping been performed as a joint call or individual by individual and then the dataset been merged?
Response: Thank you for your advice. The genotyping was performed as a joint call (CombineGVCFs module of GATK) (line 88).
Rows 84-87: how did the authors defined the filtering criteria? The hard filtering can vary greatly depending on the dataset. In absence of a truth dataset to use to train a VQSR model, it is better to assess multiple filtering values checking how the transition/transversion rate change in order to define the best set of filtering to adopt, and minimise false positives (retained) and negatives (excluded).
Response: The hard filtering is recommended by GATK. In addition, The transitions/transversions ratio was 2.3359, which was approximately equal to that in Dazu black goat (2.3347), and lower than that in IMCG goat (2.3552), suggesting that the variant calling of the samples was done correctly (lines 146-148).
In addition to the analyses performed, I would suggest to describe the number of singleton per individual, heterozygosity per individual, the Ne for the population at different generations (through SNeP or NeESTIMATOR v.2 for example) and the FROH. These additional metrics might provide additional insight into the demographic of the population, including bottlenecks and regions of unexpectedly high homozygosity.
Response: Thank you for your advice. We have added some additional metrics, including the number of singleton per individual, heterozygosity per individual and the FROH (lines 103-107). For the Ne of the population at different generations, because the softwares (SNeP or NeESTIMATOR v.2) is time-consuming for whole-genome sequence data, we don’t complete the estimation. Moreover, the demographic of the population is not our focus in this study.
Rows 109-110: why not considering also the Tajima’s D for the selection of the signals?
Response: For selective signals, if the number of algorithms is increased, the false negatives will be increased. In addition, to our knowledge, there was few reports that used Tajima’s D as the main algorithm for the selection of the signals.
Results and discussion
Rows 126-127: a coverage of 9.5 is not elevated. Please remove the adjective.
Response: Thank you for your advice. We have removed (line 142).
Row 128: number of singletons? Ts/Tv ratio for the variants selected?
Response: Thank you for your advice. We have added the information (lines 145-148).
Figure 1a and 1b: the two figures have the same caption.
Response: Thank you for your advice. We have changed (line 166).
Rows 151-152: how did you defined a consecutive region? Simply if they overlapped or if they were within a certain distance? Please add some more details.
Response: Thank you for your advice. We have changed (line 174).
Rows 195-198: the conclusion drawn by the authors is in clear contrast with the previous assumptions. If the population has higher inbreeding, less variants than the other breeds described, higher linkage decay and nucleotide diversity comparable to the other populations just means that the population has an unusually high level of homozygosity, with very few diverse haplotypes consistent with recent bottleneck. The addition of FROH, effective population size and Heterozygosity might provide better insight and allow for a deeper understanding of the population genetic diversity.
Response: Thank you for your advice. We have changed (lines 223-224).
The authors describe a series of genes under candidate selection in this population, and do not go any farther than this, despite having annotated the variants’ effect using ANNOVAR. It would be good to know whether there is any non-synonymous, high impact variants in the genes of interest.
Response: Thank you for your advice. We have added the description of the nonsynonymous SNP with the high frequency (>0.8) of alternative allele under candidate selection (lines 175-177).
How does the variants relate with other studies performed in other goat breeds? For example, the ADAPTmap consortium studied selection signatures in multiple populations worldwide, including small ones. It would be good to compare the results identified with the results produced in these studies.
Response: Thank you for your advice. We have added the comparison (lines 239-242).
Conclusions
The authors have the opportunity to already detect candidate variants that, in the next studies, could be investigated jointly with phenotypical data to detect causative variants. I think this is worth further investigation, and to be specified here too.
Response: Thank you for your advice. We have added the point (lines 277-278).
Minor Revision
Rows 12: replace cashmere with fibre.
Response: Thank you for your advice. We have changed (line 12).
Rows 96-97: specify the parameters used to perform the imputation. Have the authors retained the physically phased genotypes (PGT field in the genotype annotation by GATK) prior imputation? This because Beagle doesn’t recognise the PGT field automatically, and therefore ignores the phased genotypes. Since read-based phasing is much more reliable than statistical phasing, and therefore it is wise to retain it.
Response: Thank you for your advice. We have added the parameters used to perform the imputation and phasing (lines 112-114).
Rows 100-101: how did the author defined the ancestral alleles?
Response: Because for our data, obtaining the ancestral alleles was a problem, we used selscan which is 'dumb' with respect ancestral/derived coding and simply expects haplotype data to be coded 0/1 to calculate iHS statistic.
Rows 159-150: please provide a bit more of explanation of the meaning of the Tajima’s D negative values (excess of rare alleles, derived by recent bottleneck or selective sweep) either here or in the materials and methods
Response: Thank you for your advice. We have added the explanation (line 182).
Row 206: replace dense with defence.
Response: Thank you for your advice. We have changed (line 233).
Supplementary tables: I would suggest to break the tables in multiple, smaller pages. Not everyone has access to a A3 printer, and print these tables in A4 would make them hard to read
Response: Thank you for your advice. We have changed.

Round 2
Reviewer 1 Report
The manuscript “Identification of Genomic Characteristics and Selective Signals in a Du’an Goat flock” aims at screening for candidate genes linked to immune resistance, small body size and heat tolerance in Du’an goat used whole-genome resequencing techniques, to uncovered the unique diversity of goats, and to provide data that may help further accelerate the breeding of this breed. The methods of experiment is scientific and the organization of manuscript is logical but there are some questions to be noticed.
1 It’s better to provide the approval number of animal welfare committee of university.
2 In Table S3, how to explain that ROH of DAG32 individual was 0 ?
3 The explanations of results part 3.1 are not sufficiently refined and clear, it’s better to clarify that what meaning of genetic diversity analysis (data or figure) is.
4 Did you have the reference population to perform selection sweep analysis?
5 In discussion 4.1, it’s also not clear to clarify the genetic diversity and conservation of this breed.
6 How did you get the nonsynonymous SNPs in these candidate genes? Please specify.
7 Did you validate the influence of SNPs on phenotypes ?
Author Response
The manuscript “Identification of Genomic Characteristics and Selective Signals in a Du’an Goat flock” aims at screening for candidate genes linked to immune resistance, small body size and heat tolerance in Du’an goat used whole-genome resequencing techniques, to uncovered the unique diversity of goats, and to provide data that may help further accelerate the breeding of this breed. The methods of experiment is scientific and the organization of manuscript is logical but there are some questions to be noticed.
Response: Thank you for your careful comments.
1 It’s better to provide the approval number of animal welfare committee of university.
Response: Thank you for your advice. We have provided the approval number of animal welfare committee of university (rows 82-84).
2 In Table S3, how to explain that ROH of DAG32 individual was 0 ?
Response: We are very sorry for the mistake. The ROH of DAG32 individual was < 1000 kb (row 170). The parameter for calculating ROH (--homozyg-kb 1000) lead to the value of 0.
3 The explanations of results part 3.1 are not sufficiently refined and clear, it’s better to clarify that what meaning of genetic diversity analysis (data or figure) is.
Response: the explanations of results in part 3.1 were put in discussion. Through comparing the results of our study to previous studies, we found Du’an goat has higher SNP-based inbreeding, lower ROH-based inbreeding, less variants, higher linkage decay, smaller effective population size and nucleotide diversity, suggesting the unique genetic characteristics (rows 241-243).
4 Did you have the reference population to perform selection sweep analysis?
Response: Because we didn’t have the reference population, the method to perform selection sweep analysis was based on single population. Furthmore, a previous study was aslo based on single population to detect selective signals (doi:10.1371/journal.pgen.1004148). In addition, only 11 selected genes were identified by ADAPTmap consortium, suggesting that the majority of selective sweeps identified by this study might be specific for the popuation.
5 In discussion 4.1, it’s also not clear to clarify the genetic diversity and conservation of this breed.
Response: Thank you for your advice. We have added the description of relationship between genetic diversity and conservation of this breed (rows 233-235).
6 How did you get the nonsynonymous SNPs in these candidate genes? Please specify.
Response: Thank you for your advice. We have specified (rows 185-186).
7 Did you validate the influence of SNPs on phenotypes ?
Response: Based on the results, another study is already being planned for exploring the relationship between genotype and phenotype.
Reviewer 2 Report
Comment to the author
The revised manuscript by Chen Q et al. is a greatly improves the original manuscript, addressing most of the flaws present in the previous version. The novel analyses, including FROH and singleton analyses, definitely provides more insight into the genetic structure of the Chinese Du’an autochthonous population.
The authors addressed most of my previous concerns. Despite the major improvements to the content, the manuscript still needs a thorough English revision to address typos, poorly connected sentences and improper wording. The addition of a picture and the area of origin of the population should be expanded. Does the breed need to follow some standard? Are they all of the same colour? These pieces of information should be provided since inaccessible by international readers, and should not be limited to a small caption within the figure but properly included in the main body of the paper.
More importantly, I still have one major analytical concern to raise. The authors still haven’t performed an appropriate check for relatedness among the individuals. I agree that the sampling has been performed following good practices for selection signatures analysis. However, it is often very difficult to keep a flawless pedigree of goat breeds due to common farming practices used by farmers. As a consequence, it is not unusual the presence of underlying population structures that will strongly hamper downstream selection signature analyses, especially with limited sample sizes.
Despite adding most of the analyses requested, the authors comment only part of them. For example, the singleton analysis is completely ignored, and the missense variants are barely cited, but never commented. The latter in particular is the most relevant, since among them there might be some of great interest. For example, the goat in the picture is solid white, and among the selection signatures there is one encompassing KIT. This gene has been shown to be responsible for the solid white phenotype in several species, such as pig, mouse and horse (see for example https://doi.org/10.1111/j.1365-2052.2009.01893.x). It might be interesting to check whether there is some analogous mutation to the ones described in the other species, since it would provide better insight on the population’s genes under selection. In my opinion, this should be done for the few genes commented by the authors, especially considering the potential relevance for disease resistance.
Also, even though I understand the authors concerns on the Ne calculation, I still believe that the computation of the effective population size would help in better understanding and commenting some of the results. By instance, the lack of recent bottlenecks might suggest that the selection signatures are caused not by a recent reduction followed by expansion of the population, but instead from other selective pressures. Moreover, I can’t tell for NeESTIMATOR, but the calculation through SNeP is not that computationally intensive. We recently ran it on a WGS dataset, with roughly the same number of variants and ~more than 50 individuals. This operation took less than 24 hours with 3 cores and less than 16 G of RAM, and definitely less greedy than Beagle and GATK requirements.
Below, some additional minor revisions to the authors.
Minor Revision
- Figure 1: authors show a figure, but they never refer to that within the text. Please, link it to the appropriate point within the text. Also, details about the populations should be specified within the text, and not in the figure. Also, a proper phenotypical description should be provided, not
- Specifying the parameters is very useful to guarantee the repeatability of the analyses, I thank the author for that. However, I would suggest to make the M&M as concise as streamlined and concise as possible, and move the analytical details in the supplementary materials. In this way, people interested can simply go and check them, whereas the other readers can focus on the general experimental design.
- Row 114: Specify beagle version (i.e. BEAGLE 3/4/5/5.1).
- Rows 144-146: When specifying average values, it would be good to specify also the standard deviation, minimum and maximum values. This will better define the presence of outliers that might skew the results, especially the selection signature analyses.
- Row 174: replace ‘overlapped’ with ‘overlapping’
- Rows 175-177: why only SNPs? Indels are more likely to have disruptive effect.
- Row 182: replace ‘…out of the…’ with ‘indicative of an’. Also, add at the end the possible causes behind the increase of rare alleles, such as recent bottleneck followed by expansion.
- Row 223: I would specify that the different number of variants in the different datasets can be related to the different filtering. For example, the Moroccan goats paper performed a VQSR instead of using the hard filters. Depending on how this has been done, the filtering might have been more or less stringent.
- Row 224: replace ‘other populations’ with ‘other locally adapted populations’
Author Response
The revised manuscript by Chen Q et al. is a greatly improves the original manuscript, addressing most of the flaws present in the previous version. The novel analyses, including FROH and singleton analyses, definitely provides more insight into the genetic structure of the Chinese Du’an autochthonous population.
Response: Thank you for your careful comments. We clarified all critical points suggested by you.
The authors addressed most of my previous concerns. Despite the major improvements to the content, the manuscript still needs a thorough English revision to address typos, poorly connected sentences and improper wording. The addition of a picture and the area of origin of the population should be expanded. Does the breed need to follow some standard? Are they all of the same colour? These pieces of information should be provided since inaccessible by international readers, and should not be limited to a small caption within the figure but properly included in the main body of the paper.
Response: Thank you for your advice. We have rephrased some sentences for a thorough English revision. The selected individuals were from a breeding farm. We are very sorry for not collecting any phenotypical data. Du’an goats exhibit diverse types of coat color phenotypes (white, black and black pied). Some pieces of information were included in the main body of the manuscript (lines 236+254+267).
More importantly, I still have one major analytical concern to raise. The authors still haven’t performed an appropriate check for relatedness among the individuals. I agree that the sampling has been performed following good practices for selection signatures analysis. However, it is often very difficult to keep a flawless pedigree of goat breeds due to common farming practices used by farmers. As a consequence, it is not unusual the presence of underlying population structures that will strongly hamper downstream selection signature analyses, especially with limited sample sizes.
Response: Thank you for your advice. To check for the relatedness among the individuals, identity-by-descent testing was performed using PLINK. Only one pairs from the individuals showed PI_HAT value of greater than 0.25 (0.29), suggesting that the sampling of the Du’an goats correctly reflects reality (lines 94+152-154).
Despite adding most of the analyses requested, the authors comment only part of them. For example, the singleton analysis is completely ignored, and the missense variants are barely cited, but never commented. The latter in particular is the most relevant, since among them there might be some of great interest. For example, the goat in the picture is solid white, and among the selection signatures there is one encompassing KIT. This gene has been shown to be responsible for the solid white phenotype in several species, such as pig, mouse and horse (see for example https://doi.org/10.1111/j.1365-2052.2009.01893.x). It might be interesting to check whether there is some analogous mutation to the ones described in the other species, since it would provide better insight on the population’s genes under selection. In my opinion, this should be done for the few genes commented by the authors, especially considering the potential relevance for disease resistance.
Response: Thank you for your advice. We have added some comments for missense variants (lines 247-248+259-260+269+277-278). KIT is a famous gene for coat color, however, Du’an goats exhibit diverse types of coat color phenotypes (white, black and black pied). Therefore, we didn’t discuss the point suggested by you. For singleton analysis, there was no studies performed in other goat breeds, thus we could not find any comparisons and make any comments.
Also, even though I understand the authors concerns on the Ne calculation, I still believe that the computation of the effective population size would help in better understanding and commenting some of the results. By instance, the lack of recent bottlenecks might suggest that the selection signatures are caused not by a recent reduction followed by expansion of the population, but instead from other selective pressures. Moreover, I can’t tell for NeESTIMATOR, but the calculation through SNeP is not that computationally intensive. We recently ran it on a WGS dataset, with roughly the same number of variants and ~more than 50 individuals. This operation took less than 24 hours with 3 cores and less than 16 G of RAM, and definitely less greedy than Beagle and GATK requirements.
Below, some additional minor revisions to the authors.
Response: Thank you for your advice. We have added the Ne calculation (lines 101-102+160-161+221-224).
Minor Revision
Figure 1: authors show a figure, but they never refer to that within the text. Please, link it to the appropriate point within the text. Also, details about the populations should be specified within the text, and not in the figure. Also, a proper phenotypical description should be provided, not
Response: Thank you for your advice. The Figure 1 has been referred in line 44 within the text. The details about the populations have been specified within the text (lines 236+254+267).
Specifying the parameters is very useful to guarantee the repeatability of the analyses, I thank the author for that. However, I would suggest to make the M&M as concise as streamlined and concise as possible, and move the analytical details in the supplementary materials. In this way, people interested can simply go and check them, whereas the other readers can focus on the general experimental design.
Response: Thank you for your advice. We have moved the analytical details in the supplementary materials.
Row 114: Specify beagle version (i.e. BEAGLE 3/4/5/5.1).
Response: Thank you for your advice. We have specified beagle version (line 107).
Rows 144-146: When specifying average values, it would be good to specify also the standard deviation, minimum and maximum values. This will better define the presence of outliers that might skew the results, especially the selection signature analyses.
Response: Thank you for your advice. We have specified the minimum and maximum values (lines 140-141).
Row 174: replace ‘overlapped’ with ‘overlapping’
Response: Thank you for your advice. We have changed (line 173).
Rows 175-177: why only SNPs? Indels are more likely to have disruptive effect.
Response: Thank you for your advice. We have added (line 176).
Row 182: replace ‘…out of the…’ with ‘indicative of an’. Also, add at the end the possible causes behind the increase of rare alleles, such as recent bottleneck followed by expansion.
Response: Thank you for your advice. We have changed (line 181).
Row 223: I would specify that the different number of variants in the different datasets can be related to the different filtering. For example, the Moroccan goats paper performed a VQSR instead of using the hard filters. Depending on how this has been done, the filtering might have been more or less stringent.
Response: Thank you for your advice. We have added (line 214).
Row 224: replace ‘other populations’ with ‘other locally adapted populations’
Response: Thank you for your advice. We have changed (line 227).

Round 3
Reviewer 2 Report
Comment to the author
The revised manuscript by Chen Q et al. partially addresses the issues raised in the previous version of the paper. Adding a relatedness analysis allows better understanding and evaluation of the analyses performed.
Despite what is stated by the authors, the manuscript still needs an English revision. I’ve written below some typos and improper wording, some of which present since the very first version of the manuscript (framing instead of farming). There are far too many to simply ignore, and I would still suggest to go through a thorough English revision.
Also, as specified last time, the description of the population is a key point in the paper. It’s ok to add some phenotypical description in the results and/or discussion sections to help in understanding the conclusions. However, few sentences scattered across multiple parts of the paper or embedded in the figure do not make up for the lack of phenotypical description in the introduction. Also, according to the description of the authors in their replies, the goat in Figure 1 is not representative of the population’s variability, therefore the figure should be changed with something actually representative of the breed, and further highlighting the need of a proper description in the introduction.
The relatedness analyses highlighted the presence of a couple of related individuals. Keeping them in the subsequent analyses might hamper the results of the selection signature, especially with as few as 15 samples involved. The relatedness filtering prior to selection signature discovery is a major step, and I wouldn’t rely on any results obtained without it, especially with a sample of limited size.
Regarding the effective population size, the authors missed the point that I raised last time. The Ne should be used to better comment the selection signatures, to better understanding the origin of the selection signatures identified. By instance, it shows the likely absence of bottlenecks in recent generations for this specific population. This allows the authors to better comment on the signatures detected, making bottlenecks a less likely cause of the reduction in variability and sweeps identified.
Last but not least, the fact that all samples are originating from a single farm makes this characterisation less representative of the population. I think that the authors should clearly state that this is a pilot study, which will need a broader and more thorough investigation of the diversity.
Typesetting
- Row 14: replace `follow-up analysis` with `follow-up analyses`
- Row 15: replace utilized with analyzed/investigated
- Row 17: Not having phenotypic data, you can’t tell whether the regions are responsible for any particular phenotype. I would simply say `and candidate genomic regions which might be involved in adaptive traits`
- Row 23: `underlying its adaptive traits`
- Row 26: have you compared the Ne with other breeds to check whether it is smaller/larger/comparable? If so, it should be specified in the main body too.
- Row 30: unique diversity, without adaptive
- Row 37: farming, not framing.
- Row 38: replace cashmere with fiber.
- Row 39: replace `they now comprise` with `there are now`
- Row 41: replace `distributed` with `reared`
- Row 42: replace `sporadically distributed` with `sporadically found`
- Row 44: disease resistance is too vague. Please clarify the disease resistance and expand this description: limit it to the figure is not enough.
Etc..
Author Response
The revised manuscript by Chen Q et al. partially addresses the issues raised in the previous version of the paper. Adding a relatedness analysis allows better understanding and evaluation of the analyses performed.
Despite what is stated by the authors, the manuscript still needs an English revision. I’ve written below some typos and improper wording, some of which present since the very first version of the manuscript (framing instead of farming). There are far too many to simply ignore, and I would still suggest to go through a thorough English revision.
Response: Thank you for your careful comments. The manuscript have been gone through a English revision.
Also, as specified last time, the description of the population is a key point in the paper. It’s ok to add some phenotypical description in the results and/or discussion sections to help in understanding the conclusions. However, few sentences scattered across multiple parts of the paper or embedded in the figure do not make up for the lack of phenotypical description in the introduction. Also, according to the description of the authors in their replies, the goat in Figure 1 is not representative of the population’s variability, therefore the figure should be changed with something actually representative of the breed, and further highlighting the need of a proper description in the introduction.
Response: Thank you for your advice. We have deleted the goat in Figure 1 and added phenotypical description in the introduction (rows 51-52).
The relatedness analyses highlighted the presence of a couple of related individuals. Keeping them in the subsequent analyses might hamper the results of the selection signature, especially with as few as 15 samples involved. The relatedness filtering prior to selection signature discovery is a major step, and I wouldn’t rely on any results obtained without it, especially with a sample of limited size.
Response: Thank you for your advice. We have removed one individual which was related with another individual (identity-by-descent PI_HAT=0.29) for selective sweep analysis (rows 112-113).
Regarding the effective population size, the authors missed the point that I raised last time. The Ne should be used to better comment the selection signatures, to better understanding the origin of the selection signatures identified. By instance, it shows the likely absence of bottlenecks in recent generations for this specific population. This allows the authors to better comment on the signatures detected, making bottlenecks a less likely cause of the reduction in variability and sweeps identified.
Response: Thank you for your advice. We have added the point (row 232).
Last but not least, the fact that all samples are originating from a single farm makes this characterisation less representative of the population. I think that the authors should clearly state that this is a pilot study, which will need a broader and more thorough investigation of the diversity.
Response: Thank you for your advice. We have stated (rows 294-296).
Typesetting
Row 14: replace `follow-up analysis` with `follow-up analyses`
Response: Thank you for your advice. We have changed (row 19).
Row 15: replace utilized with analyzed/investigated
Response: Thank you for your advice. We have changed (row 21).
Row 17: Not having phenotypic data, you can’t tell whether the regions are responsible for any particular phenotype. I would simply say `and candidate genomic regions which might be involved in adaptive traits`
Response: Thank you for your advice. We have changed (row 22).
Row 23: `underlying its adaptive traits`
Response: Thank you for your advice. We have changed (row 29).
Row 26: have you compared the Ne with other breeds to check whether it is smaller/larger/comparable? If so, it should be specified in the main body too.
Response: Thank you for your advice. We have specified the point (row 31-33).
Row 30: unique diversity, without adaptive
Response: Thank you for your advice. We have changed (row 37).
Row 37: farming, not framing.
Response: Thank you for your advice. We have changed (row 44).
Row 38: replace cashmere with fiber.
Response: Thank you for your advice. We have changed (row 45).
Row 39: replace `they now comprise` with `there are now`
Response: Thank you for your advice. We have changed (row 46).
Row 41: replace `distributed` with `reared`
Response: Thank you for your advice. We have changed (row 48).
Row 42: replace `sporadically distributed` with `sporadically found`
Response: Thank you for your advice. We have changed (row 49).
Row 44: disease resistance is too vague. Please clarify the disease resistance and expand this description: limit it to the figure is not enough.
Response: Thank you for your advice. We have expand the description (row 52).
Etc..
Response: Thank you for your advice. We have rephrased a lot of sentences (row s 204+232+260).

Round 4
Reviewer 2 Report
Comment to the author
The revised manuscript by Chen Q et al. addresses the scientific issues raised in the previous version of the paper. Although greatly improved, the manuscript still needs an appropriate English revision, considering the presence of some sparse typos and unusual wording of the sentences.
The description of the population is still too short, since it is simply the same previously present in the figure, without expanding it as requested in the previous reviews. Please, expand it adding a proper phenotypical description (coat colour, horn presence and type, coat length, milk production values if known etc.). These pieces of information might sound trivial, but might be extremely useful for reader to better understand the population of interest. Moreover, my criticism on the goat picture was that was not representative. If a picture of a flock was available, that would be ideal to appreciate the population under investigation.
I would also stress how this should classified as a preliminary investigation. This should emerge strongly from the paper since a sample of 15 animals from one flock cannot be considered representative of such a large population. Therefore, I suggest to add this in the title and abstract.
Finally, my last observation is regarding the Ne comment. The sentence at line 232 is too strong: the Ne is not confirming the presence of selective sweep, suggesting the presence of a stronger selective pressure/recent bottleneck.
Other minor notes:
- The reference for gene LCOR should probably be 34, not 31
- The comment for MTOR is too short and doesn’t expand enough the role of the gene in heat response and adaptation. Same for ABCG2. In general, I find the paragraph poorly connected and should be rephrased to fully explain how the genes are involved in heat resistance.
Author Response
The revised manuscript by Chen Q et al. addresses the scientific issues raised in the previous version of the paper. Although greatly improved, the manuscript still needs an appropriate English revision, considering the presence of some sparse typos and unusual wording of the sentences.
Response: Thank you for your careful comments. A high-quality paper never deviates from a good reviewer.
The description of the population is still too short, since it is simply the same previously present in the figure, without expanding it as requested in the previous reviews. Please, expand it adding a proper phenotypical description (coat colour, horn presence and type, coat length, milk production values if known etc.). These pieces of information might sound trivial, but might be extremely useful for reader to better understand the population of interest. Moreover, my criticism on the goat picture was that was not representative. If a picture of a flock was available, that would be ideal to appreciate the population under investigation.
Response: Thank you for your advice. We have changed the goat picture and added some descriptions about the population (rows 51-54).
I would also stress how this should classified as a preliminary investigation. This should emerge strongly from the paper since a sample of 15 animals from one flock cannot be considered representative of such a large population. Therefore, I suggest to add this in the title and abstract.
Response: Thank you for your advice. We have added the point (rows 3+22+30).
Finally, my last observation is regarding the Ne comment. The sentence at line 232 is too strong: the Ne is not confirming the presence of selective sweep, suggesting the presence of a stronger selective pressure/recent bottleneck.
Response: Thank you for your advice. We have changed (row 234).
Other minor notes:
The reference for gene LCOR should probably be 34, not 31
Response: Thank you for your advice. We have changed (row 267).
The comment for MTOR is too short and doesn’t expand enough the role of the gene in heat response and adaptation. Same for ABCG2. In general, I find the paragraph poorly connected and should be rephrased to fully explain how the genes are involved in heat resistance.
Response: Thank you for your advice. We have added two literatures of MTOR gene (row 281-284) and one literature of ABCG2 gene (row 286).

Round 5
Reviewer 2 Report
The latest revision of the paper is acceptable in terms of content. The English is improved and, although some wordings are sentences are not always sound, I couldn't detect any typos in the text.
One last remark can be the sentence:
'The decreasing recent Ne was likely due to the presence of a stronger selective/recent bottleneck.'
Could be rephrased to something like:
'A recent decrease in Ne can be due to a stronger selective selective pressure in recent generations or to a recent bottleneck.'
Or something similar.
All the best
Author Response
The latest revision of the paper is acceptable in terms of content. The English is improved and, although some wordings are sentences are not always sound, I couldn't detect any typos in the text.
Response: Thanks to reviewer 2. Your comments help us improve our manuscript.
One last remark can be the sentence:
'The decreasing recent Ne was likely due to the presence of a stronger selective/recent bottleneck.'
Could be rephrased to something like:
'A recent decrease in Ne can be due to a stronger selective selective pressure in recent generations or to a recent bottleneck.'
Or something similar.
Response: Thank you for your advice. We have rephrased (rows 234-235).